# Phycocosmetics and Other Marine Cosmetics, Specific Cosmetics Formulated Using Marine Resources

**DOI:** 10.3390/md18060322

**Published:** 2020-06-18

**Authors:** Céline Couteau, Laurence Coiffard

**Affiliations:** Laboratoire de Pharmacie industrielle et de Cosmétologie, Faculté de Pharmacie, Université de Nantes, MMS, EA2160, 61112–44035 Nantes, France; celine.couteau@univ-nantes.fr

**Keywords:** algae, substance of interest, cosmetic products

## Abstract

Marine resources exist in vast numbers and show enormous diversity. As a result, there are likely many possible applications for marine molecules of interest in the cosmetic industry, whether as excipients or additives, but especially as active substances. It is possible to obtain extracts from active substances; for example, quite a few algae species can be used in moisturizing or anti-ageing products. In the field of topical photoprotection, mycosporine-like amino acids and gadusol are important lines of enquiry that should not be overlooked. In the field of additives, the demonstration that certain seaweed (algae) extracts have antimicrobial properties suggests that they could provide alternatives to currently authorized preservatives. These promising leads must be explored, but it should be kept in mind that it is a long process to bring ingredients to market that are both effective and safe to use.

## 1. Introduction

According to the Statista Institute, the value of retail sales of cosmetic products on the European market, which has been steadily growing over the past five years, totaled 77.6 billion Euros in 2017. This makes Europe the world’s largest market for this type of product [1,2]. Skin care and personal care products remain the main categories of cosmetics in Europe, followed by hair products and perfumes [2]. In the European Union, at least 5795 companies manufacture cosmetic products, and just over 1000 of these companies are found in France [1]. The major groups, L’Oréal, Chanel and Guerlain, occasionally use marine raw materials in their products, whereas certain much smaller companies, located primarily in Brittany, such as Bretagne Cosmétiques Marins, Phytomer or Seamer, base the whole of their marketing on the fact that their products are formulated with substances from the sea. Capitalizing on the growing need to attract consumers through the use of natural or organic products, “blue technology” is a way to obtain new original ingredients. The marine ecosystem is very rich and numerous species of macroalgae, microalgae and halophilic plants may be used to formulate products with varied properties (moisturizing, anti-ageing, slimming, etc.), which we will discuss here.

## 2. Use of Marine Resources in Cosmetology

In the European Union, a cosmetic product is defined in Article 2 of Chapter I of Regulation (EC) no. 1223/2009 as ‘any substance or mixture intended to be placed in contact with the external parts of the human body (epidermis, hair system, nails, lips and external genital organs) or with the teeth and the mucous membranes of the oral cavity with a view exclusively or mainly to cleaning them, perfuming them, changing their appearance, protecting them, keeping them in good condition or correcting body odors [3].

It is also possible to define a cosmetic product using a galenic formula, by taking the various categories of ingredients that make it up into account, i.e.,
1 cosmetic product = ∑Excipients +/− Active substance(s) +/− Additive(s)
where the active substance is the ingredient responsible for the claimed cosmetic activity (i.e., the equivalence of a drug’s active principle) and the additives are the substances added to preserve the product and to improve its organoleptic properties. The excipient ensures that the desired pharmaceutical dosage form is obtained [4].

### 2.1. The Marine World, a Source of Excipients

Given that very few cosmetics are anhydrous (only powders, lipsticks and nail polishes), the universal excipient is water, and sea water can be used provided that sufficient microbiological and chemical quality can be ensured. Seawater is well-known as a source of minerals that can have benefits in quite a few skin disorders, particularly inflammatory dermatitis such as atopic eczema, thus echoing the suggestions of René Quinton in the early 20th century [5,6,7]. It could even play a role in photoinduced skin cancer prevention [8]. In most cases, it is not possible for users to know where the seawater used in the cosmetic product came from as it is simply listed by its INCI name *Aqua maris*. The origin of this sea water is only known in a few cases; one of these is Dead Sea water. This water is characterized by a salinity of 345 g of minerals per liter, i.e., approximately 7 to 10 times higher than that found in oceans. Sea water is composed of chlorides (212.4 g/l), magnesium (40.65 g/l), sodium (39.15 g/l), calcium (16.86 g/l), potassium (7.26 g/l), bromides (5.12 g/l), sulfates (0.47 g/l) and bicarbonate (0.22 g/l). It has been proven that balneotherapy with Dead Sea water combined with phototherapy is an effective treatment for several types of psoriasis [9]. In cosmetic products, Dead Sea water is presented as a natural humectant, but it is also claimed to have protective effects against UVB-induced stress [10,11]. However, there appears to be a certain level of toxicity related to the minerals present in products used in both balneotherapy and mud therapy [9]. Nickel and chromium have been detected in Dead Sea mud and therefore its use is not recommended for people who are known to be sensitive to these substances. It is not surprising to find traces of nickel and chromium since we know that nickel is naturally present in volcanic soils and dust and that chromium is an element present in rocks, soils, volcanic dust and gases, as well as in the animal kingdom and in plants [12,13]. Specifically, Cr^6+^ is produced when implementing certain industrial processes and its potential increase, if it is detected in mud from the Dead Sea, may be an indication of recurrent pollution [14]. In accordance with European regulations, it is the manufacturer’s responsibility to ensure that the raw material used does not endanger the end user. We, therefore, need to be very cautious with this type of resource.

### 2.2. The Marine World, a Source of Active Substances

#### 2.2.1. The Marine World, a Source of Moisturizing Agents

One of the main functions of the skin is to provide the body with effective protection from the environment by maintaining an efficient epidermal barrier, not only with regard to the outside world, but also in terms of excessive water loss. Water homoeostasis is a requirement for normal physiological skin functioning. The Natural Moisturizing Factor (NMF) present in the *Stratum corneum*, i.e., the outermost layer of the epidermis, is comprised of lactic acid, urea, pyrrolidone carboxylic acid and numerous amino acids such as serine [15,16]. However, these constituents are not the only endogenous hygroscopic factors involved in epidermal water uptake. Glycerol, long used as a cosmetic ingredient, also plays an important physiological role in *Stratum corneum* hydration [17]. Naturally present in the epidermis and *Stratum corneum*, glycerol can come from the metabolism of fat in sebaceous glands and from the conversion of phospholipids to free fatty acids (FFA), as well as from the general circulation. In the latter case, glycerol is then transported through the epidermis via specific aquaporin water/glycerol channels. Aquaporin 3 is abundant in the epidermis and its expression is stimulated by various molecules, such as retinoic acid, along with an increase in glycerol transport activity [18]. For dry or dehydrated skin, it is easy to formulate effective cosmetic products based on two principles: the use of ingredients with film-forming and occlusive properties, such as hydrocarbons (petrolatum, paraffin oil, perhydrosqualene), silicones, fatty alcohols, vegetable oils, waxes and butters, etc., and the use of humectant agents, which increase the capacity of the *Stratum corneum* to capture water (i.e., humectants such as glycerin or propylene glycol and moisturizers such as the constituents of the NMF which bind to the corneous layer, penetrate it and allow water to be retained within *Stratum corneum* cells) [4]. In this paper, we will assess the interest of several algal species that may be able to provide bioactive substances for hydration and which are likely to replace traditional active substances and/or complement their action. One example of this is *Cladophora glomerata*, a filamentous green microalga found in marine environments as well as in fresh water [19]. This species contains saturated fatty acids (palmitic acid C16:0) and, in particular, unsaturated fatty acids C16:1 (n-7) and C18:1 (n-3) [20] that may act as an emollient and protect skin from excessive water loss. Furthermore, it was possible to isolate a sulfated polysaccharide from this alga, suggesting that there may be a moisturizing effect that can be exploited. Results obtained with an emulsion containing only 0.5% of an extract obtained by supercritical CO_2_ extraction appear quite promising.

#### 2.2.2. The Marine World, a Source of Anti-Ageing Active Substances

Skin ageing is a slow and complex process due to intrinsic and extrinsic causes [21]. Over time, we observe numerous changes that will, in particular, affect the dermis and epidermis, which become thinner. The main signs are wrinkles, dryness, loss of firmness and elasticity, skin sensitivity, fine lines and wrinkles [21,22]. Intrinsic causes remain essentially genetic and it is therefore not possible to act on them. Conversely, extrinsic causes, which include smoking, poor diet, lack of sleep, stress, excessive alcohol consumption and exposure to the sun, are the only ones that can be influenced. As one of the main theories of ageing is the radical theory, anti-radical substances would be a relevant choice when formulating anti-ageing products. For this reason, it is likely that brown algae such as *Eisenia arborea* (Areschoug) or *Ecklonia cava* (Kjellman), *Laminaria japonica* or *Fucus* contain interesting raw materials [23,24,25,26,27].

The dermis is primarily made up of the extracellular matrix (ECM), which is primarily made up of specific cells called fibroblasts. During the ageing process, the dermis undergoes significant changes. Collagen, which is one of the major components of the ECM, breaks down and its total amount decreases.

This is linked to increased activity of matrix metalloproteinases (MMPs) [28]. Thus, it appears that the use of substances capable of inhibiting these metalloproteinases is definitely relevant in an anti-ageing product. Sulphated polysaccharides found in Rhodophyceae, Phaeophyceae and Chlorophyceae, as well as polyphenols, derived from phloroglucinol, have revealed properties of this type [29]. For example, very interesting results have been obtained with ethanolic extracts from *Jania rubens*, which have a significant effect on elastase, hyaluronidase and metalloproteinase MMP-1 [30].

Progerin also seems to be an avenue to explore. Current data in the literature indicate that progerin, which is a mutant of lamin A, may be one of several previously known physiological biomarkers of the ageing process, which begins at the age of 30 years [31,32,33]. Progerin accumulates as the skin ages, so its expression is higher in young fibroblasts [34]. An extract of *Alaria esculenta*, an edible seaweed, has been shown to result in a significant decrease in the amount of progerin in older fibroblasts at the lowest concentration tested [35].

#### 2.2.3. The Marine World, a Source of Soothing Active Substances

Under the term ’soothing effect’, we find ingredients that have demonstrated anti-inflammatory activity. For example, in folk medicine, marine algae such as *Sargassum* have been used for these purposes [36], as have *Porphyra* species [37]. Phenolic compounds, well-known for their numerous biological functions, should be mentioned again here [38]. In particular, brown algae (*Ecklonia cava*, *Eisenia arborea*, *Ecklonia stolinifera* and *Eisenia bicyclis*) have been studied for this reason, as they are interesting sources of bioactive phenolic compounds [39]. Phlorotannins from brown algae exhibited anti-inflammatory effects on mouse ear edema and are considered particularly potent inhibitors of proinflammatory cytokines such as nitric oxide synthase (iNOS), cyclooxygenase-2 (COX-2), tumor necrosis factor alpha (TNF-α) and interleukin-1 beta (IL-1β) and 6 (IL-6) [40].

#### 2.2.4. The Marine World, a Source of Slimming Active Substances

The use of algae in the field of slimming is surely their earliest known use. The term slimming product is always used in cosmetology, and never anti-cellulite product, since cellulite or lipodystrophy is a disorder caused by a modification of the deep dermis and subcutaneous tissue, which results in a padded appearance on the surface [41]. The hypodermis contains adipocytes for which the vacuolar volume will significantly increase, causing the cell to reach an exaggerated size [42]. The main lipolytic substances used in cosmetology are xanthic bases, especially caffeine and algae rich in iodine and, to a lesser extent, L-carnitine, coenzyme A and plant-derived active ingredients such as forskolin or escin [4]. The justification for the use of iodine-rich algae is related to the fact that iodine is involved in the synthesis of thyroid hormones, which promote lipolysis by facilitating the penetration of fatty acids into the mitochondria due to the increased synthesis of palmitoylcarnitine [43,44]. Since algae are able to concentrate iodine from seawater, some species are naturally rich in this element. One example of this is *Laminaria Japonica*, which may contain up to 0.9% iodine in terms of dry weight [45]. Depending on the species and where they are harvested, iodine concentrations may vary widely [46]. In any case, its use creates a regulatory paradox since iodine is on the list of ingredients prohibited in cosmetic products given in Annex II of Regulation (EC) no. 1223/2009.

#### 2.2.5. The Marine World, a Source of Substances for Protection against UV Radiation

Exposure to ultraviolet radiation results in actinic erythema or ‘sunburn’ over the short term and immunosuppression, premature skin ageing and skin cancer over the long term [47]. It is therefore important to protect oneself from UV radiation using appropriate clothing, as well as sunscreen, the use of which is an integral part of an overall prevention strategy [48,49]. Approximately twenty organic filters are available for their formulation and two mineral ones: titanium dioxide and zinc oxide [50]. Some of these filters are controversial due to their sensitizing properties or potential endocrine disrupting effect [51,52,53]. It is also alleged that they are responsible for coral bleaching, although this phenomenon obviously depends on multiple factors. Without a doubt, the filter most often accused of this bleaching is oxybenzone, a broad-spectrum filter from the benzophenone family [54,55]. This means it is necessary to find molecules of interest for photoprotection from the marine world that may be able to replace the existing ones. Mycosporine-like amino acids (MAAs) are unique, low-molecular weight, water-soluble molecules with a maximum absorption between 310 and 360 nm. These molecules are found in many marine resources such as phytoplankton, sea fans, sponges, shrimp, sea urchins, starfish, clams and algae, although most MAA-producing marine algae are red algae [56,57]. Unlike photosynthetic pigments, it has been suggested that MAAs behave like passive protective substances, dissipating the energy of incident radiation absorbed in the form of heat, without generating reactive toxic oxygen species [58,59]. A mixture of MAAs is already marketed under the name Helioguard^®^365 by the Mibelle Biochemistry group. This cosmetic ingredient combines shinorine (Figure 1) and porphyra-334 (Figure 1), extracted from the red alga *Porphyra umbilicalis*.

Other Rhodophyceae such as *Agarophyton vermiculophyllum*, *Crassiphycus corneus*, *Gracilariopsis longissima* and *Pyropia leucostica* should be able to provide interesting sources of these MAAs [60].

Another interesting track to explore is that of the use of gadusol (Figure 2), a molecule that is structurally related to mycosporines and which was isolated for the first time from cod eggs (*Gadus morhua* L.), which contain approximately 4 g/kg dry weight.

Gadusol has been identified as 1,4,5-trihydroxy-5-hydroxymethyl-2-methoxycyclo-hex-en-3-one [61,62] and has similar antioxidant properties to ascorbic acid [61]. With an implementation mechanism comparable to that of MAAs, gadusol may be a very promising substance for topical photoprotection [63]. The effectiveness of gadusol, as well as of the various MAAs, still needs to be tested via the near-universal indicator, the Sun Protection Factor (SPF), so it can be compared with the various filters that are currently authorized.

One effect of UV irradiation is the appearance of skin spots [64,65]. These unsightly spots are removed through the use of substances that are capable of inhibiting tyrosinase, which is the key enzyme in melanogenesis. Marine algae have recently attracted attention in the search for natural tyrosinase inhibitors. On this basis, fucoxanthin, isolated from *Laminaria japonica*, has demonstrated an ability to inhibit tyrosinase activity in UVB-irradiated guinea pigs and melanogenesis in UVB-irradiated mice [66]. Phloroglucinol derivatives, another common secondary metabolite found in brown algae, also have inhibitory activity against tyrosinase due to their ability to chelate copper [67]. Although these substances appear to be good candidates in in vitro and in vivo experiments in animals, an in vivo confirmation in volunteers with lentigo still needs to be carried out.

### 2.3. The Marine World, a Source of Additives

The term additives refers to all substances used to improve the preservation of cosmetic products, as well as their organoleptic characteristics, i.e., their odor and color. It is extremely important to find new ingredients to be used as additives because many of the currently used ingredients cause dermatological problems and/or are controversial. For this reason, the search is on for substitutes free of undesirable effects.

#### 2.3.1. The Marine World, a Source of Gel Forming Agents

Alginates and carrageenans are the primary marine substances used in cosmetology as additives, although their use is very limited compared to what is practiced in the food industry [68].

Alginates (Figure 3) are polysaccharides. They are compounds (1,4-polyuronic) containing three types of block structure: M block (β-D-mannuronic acid), G block (poly α-L-guluronic acid), and MG block (containing both polyuronic acids), naturally present in the cell wall and extracellular matrix of various species of brown algae or Phaeophyceae and which can represent up to 40% of the biomass.

The use of alginates is linked to their physical properties and their excellent biocompatibility and biodegradability [68,69]. However, their use is limited by their method of gelation, which requires the presence of a divalent cation, most often the Ca^2+^ ion [70], which is not advisable in a number of cosmetic forms. Therefore, they are currently almost exclusively used in mask-gel formulations for professional use only in beauty salons.

Carrageenans are linear polysaccharides with a high molecular weight containing sulfated and non-sulfated galactose and 3,6-anhydrogalactose (3,6 AG) units, linked by alternating α-(1,4) and β-(1,3) linkages (Figure 4), present in red algae or Rhodophyceae [71,72].

Historically, *Chondrus crispus* was the first species used for carrageenan extraction due to its high concentrations of this type of biopolymer (close to 55% of the biomass) [73]. The viscosity of the gel obtained depends on the gelling agent concentration, temperature, presence of other solutes and type of carrageenan used, as well as its molecular weight [74]. Different types of carrageenans exist based on the number and position of the sulfate groups, as well as the number of 3,6-anhydro-galactose (3,6 AG) bridges. The main carrageenan used is Kappa-carrageenan; this form is almost exclusively used in toothpaste formulas [75].

Chitosan (Figure 5) is another interesting biopolymer for the cosmetic industry. It is comprised of β-(1,4)-2-acetamide-D-glucose units linked to β-(1,4)-2-amino-D-glucose residues [76].

Chitosan is produced by chemical or enzymatic deacetylation of chitin, a component found in crustacean exoskeletons (crab, shrimp, lobster, krill, crayfish, barnacles) and arthropods or cephalopod endoskeletons or fungal walls [77,78,79]. Chitosan can be extracted from the cell wall of certain fungi in the Mucorales family in fewer steps and using less solvent than when deacetylating chitin extracted from crustaceans, and in a way that is considered a sustainable environmental technology. It is used in two main fields: oral hygiene products and hair products, applications which benefit from this substance’s specific properties. Studies suggest that chitosan has a broad-spectrum antimicrobial activity, that it is able to interfere with the adhesive properties of cariogenic bacteria and colonization of the oral cavity and, like fluoride, can prevent the demineralization of teeth and the formation of dental biofilms [79,80,81]. In addition, chitosan has the ability to interact with keratin from the hair, thereby exerting a conditioning effect. This makes hair fibers soft and protects them from external aggressions; it is also used in so-called volumizing shampoos [78].

#### 2.3.2. The Marine World, a Source of Preservatives

Numerous substances present in algae have been demonstrated to have antimicrobial properties. We will only discuss proteins, polysaccharides and polyphenols here [82,83,84,85,86,87].

The most effective marine alga against the bacteria tested, i.e., *Escherichia coli* (ATCC 25322), *Pseudomonas aeruginosa* (ATCC 27853), *Staphylococcus aureus* (ATCC 29213) and *Enterococcus faecalis* (ATCC 29212), was *Gracilaria dendroides*, an alga with a protein content of 13.4%. Its antimicrobial efficacy is directly related to this protein content because *Ulva reticulata*, *Cladophora socialis* and *Cladophora occidentalis*, with values of 5.8, 2.3 and 1.7%, respectively, were nowhere near as effective against the bacteria [88]. Studies indicate that sulfated polysaccharides, particularly alginates, fucoidans and laminaranes, extracted from seaweed are effective against *Escherichia coli* and *Staphylococcus aureus* [89].

Polyphenols also have antimicrobial properties. Rutin, quercetin and flavonoids extracted from kaempferol have been identified in all species of algae with antimicrobial potential. According to the results published by Al-Saif et al. (2014), the alga *G. dendroides* had the highest concentration of these three flavonoids (rutin, 10.5 mg/kg; quercetin, 7.5 mg/kg; kaempferol, 15.2 mg/kg) associated with the inhibition of *E. coli*, *P. aeruginosa*, *S. aureus* and *E. faecalis.* Forty compounds have been identified in the essential oil of the red alga *Jania rubens,* including hydrocarbons (n-docosane, n-eiscosane, n-tetratriacontane, 1-heptadecanamine, n-hexadecane, n-octadecanol, n-octadecananol, n-contatriane) found in other species [90,91,92]. Different brominated diterpenes of the parguerene and isoparguerene series [93] have been isolated using methanol extraction and it is precisely the methanolic extracts that demonstrate the greatest antimicrobial effects [90]. However, the cosmetic use of these derivatives is limited by their potential cytotoxicity [93] and a ban on the solvent.

#### 2.3.3. The Marine World, a Source of Odorous Molecules for Use in Perfumes

Over 35,000 species of algae are capable of releasing odors in aquatic environments, although few of them have been characterized in terms of their effects on human health [94]. The type of odor varies depending on the molecules produced by the various taxa [95,96]. The main odorous components produced by algae and cyanobacteria are terpenoids, carotenoids, fatty acid derivatives and sulfur compounds [97]. Some of the closely studied odorous compounds are geosmin or trans-1, 10-dimethyl-trans-9-decalol (Figure 6) and 2-methylisobornéol (MIB) with earthy/musty odors [98,99,100,101].

We know that geosmin is produced by various species of cyanobacteria such as *Oscillatoria*, *Lyngbya*, *Symploca* and *Anabaena* [102]. It has been shown that geosmin is formed by the cyclisation of farnesyl diphosphate, during which the reaction is catalyzed by geosmin synthase in cyanobacteria [103]. β-cyclocitral (Figure 7), a substance in the norcarotenoid group with a tobacco odor, as well as β-ionone, with a woody and fruity odor, have been reported as being produced by the green alga *Ulothrix fimbriata* [104].

Lastly, a number of polyunsaturated fatty acid derivatives (PUFA) extracted from algae are considered as strong odoriferous substances characterized by a fishy, rancid or cucumber odor [105,106]. The cucumber odor produced by *Synura*, for example, is attributed to 2,6-nonadienal [107]. Filamentous cyanobacteria (*Calothrix*, *Plectonema*, *Phormidium* sp. and *Rivularia* sp.) are capable of producing numerous odorant PUFA derivatives, including 1,3,3-trimethyl-2,7-dioxabicyclo (2,2,1) heptane (TDH) and 6-methyl-5-hepten-2-one (Höckelmann and Jüttner, 2005), 1-penten-3-one, 1-penten-3-ol, 2(*Z*)-pentenal, 2(*E*)-pentenal, 2(*E*),4(Z)-heptadienal and 2(*E*),4(*E*)-heptadienal [104]. Furthermore, diatoms such as *Asterionella formosa, Achnanthes minutissima, Amphora pediculus, Cymbella minuta* and *Gomphonema angustum* are capable of excreting 2(E),4(Z)-heptadienal, 2(E),4(Z)-octadienal, octa-1,5-dien-3-ol, 1,3(E),5(Z)-octatriene and ectocarpene [108,109].

#### 2.3.4. The Marine World, a Source of Dyes

The dyes currently being used in the cosmetic industry are primarily synthetic; other dyes come from minerals such as oxides and to a lesser extent, due to stability issues, from plants. The industry is searching for new dyes as some of those currently used cause allergies and some consumers want cosmetics formulated with more naturally occurring substances.

It is likely that the marine world will be able to supply the cosmetics industry with both phycobilins and carotenoids, which means that a wide range of blue, yellow, orange and red colors will be covered.

The tetrapyrroles called phycobilins represent the main photosynthetic accessory pigments of certain cyanobacteria and eukaryotic algae belonging to the Glaucophyte, Crystophyte and Rhodophyte groups. These pigments are covalently bound proteins, called phycobiliproteins, which are, in general, organized into phycobilisomes on thylakoid membranes. They are classified into four categories: phycoerythrins, phycocyanins, phycoerythrocyanins and allophycocyanins [110]. As they are all water-soluble molecules, their use in cosmetology will be limited to modifying organoleptic properties, as it is not feasible to use them in make-up. R-Phycoerythrin (R-PE) (Figure 8) is certainly one of the best studied phycobiliproteins.

R-phycoerythrin is extracted from red microalgae and has numerous possible applications, primarily in the field of immunodiagnostics [111]. An alga such as *Portieria hornemannii* (Lyngbye) Silva appears to be a good candidate for R-PE production given that a high-quality molecule can be extracted and high yields obtained. *Corallina officinalis* is also an interesting alga for anyone who wants to extract high-quality R-PE, which is the general case for use in health products, and more specifically for use in the cosmetic products of interest to us here. It should be remembered that extraction is a long and tedious procedure that requires three successive purification phases [112]. R-PE is also commonly extracted from *Ceramium isogonum* [113], *Gracilaria longa* [114], *Gracilaria fisheri* [115] and *Palmaria palmata* [116]. Perhaps it is more interesting on an industrial level to consider cyanobacteria as a source of pigments because of their rich content [117]. Various cyanobacteria species have been reported to be sources of phycocyanin, such as *Arthrospira* (*Spirulina*) *plantesis*, *Arthrospira* (*Spirulina*) *maxima*, *Pyrophyridium* sp. and *Synechocystis* [118].

Cyanobacteria are also sources of carotenoids, the second major family of dyes useful in cosmetics. The main carotenoids are β-carotene, astaxanthin, lutein, lycopene, canthaxanthin and fucoxanthin [119]. The microalga strain *Haematococcus pluvialis* (Chlorophyceae, Volvocales) is considered to be the best natural source of astaxanthin (Figure 9) [120,121].

## 3. Stringent Monitoring Requirements

Cosmetic products must not cause damage to human health when applied under normal or foreseeable conditions for use. For marine raw materials, it is necessary to know a lot of information, such as the name of the alga from which the ingredient is derived, the method used for extraction, the main composition of the extract, the identity of any preservatives, additives or contaminant and microbiological quality. Raw materials from the sea, like any raw material, must be rigorously selected and tested so as to ensure that prohibited substances such as organic pollutants or heavy metals are not introduced into the cosmetic product [122].

### 3.1. Organic Pollutants in Marine Raw Materials

Marine pollution caused by accidental spills is a well-known global concern. Oil spills are also an importance source of VOCs (Volatile organic compounds) such as various alkanes and benzene-toluene-ethylbenzene-xylene isomers and other lighter substituted benzene compounds [123]. On the other hand, it has been reported that herbicides and insecticides have been found in large doses. Metribuzin DADK, for example, has been identified and assayed in *Cystoseira corniculata* (5.01 mg/kg), *Corallina elongata* (0.703 mg/kg) and *Jania rubens* (3.85 mg/kg). Molinate was also considered as a minor contaminant found only in *Corallina elongata* (0.002 mg/kg), and pyrethrin I was identified and measured only in *Padina pavonica* and *Corallina elongata* at doses of 1.214 and 0.229 mg/kg, respectively [124]. These examples show the importance of analytical testing of raw materials.

### 3.2. Heavy Metals and Radionuclides

A study of the United States Environmental Protection Agency revealed the presence of various inorganic pollutants (copper, arsenic, aluminum, iron, manganese, cadmium, lead and zinc) in the marine environment that may have the potential to pose a risk to human health [125]. Measurements made of artificial radionuclides released into the marine environment did reveal that radionuclides are concentrated by marine biological species [126].

Optimal quality of the raw materials is fundamental to ensure consumer safety.

## 4. Conclusions

We estimate that there are several million species of algae in the world. This makes it possible to envisage many possible applications for molecules of interest in the cosmetic industry, whether as excipients or additives, but especially as active substances. It should always be kept in mind that just because these substances occur naturally this does not mean that an in-depth toxicological study does not need to be carried out prior to suggesting their use in concrete applications. Stability studies should also be carried out. As a result, there is a long way to go between demonstrating any kind of in vitro activity and large-scale use by industry in compliance with regulations such as REACH (a European Union regulation entitled ‘Registration, Evaluation, Authorisation and Restriction of Chemicals’) and regulations more specific to cosmetics.

## Figures and Tables

**Figure 1 marinedrugs-18-00322-f001:**
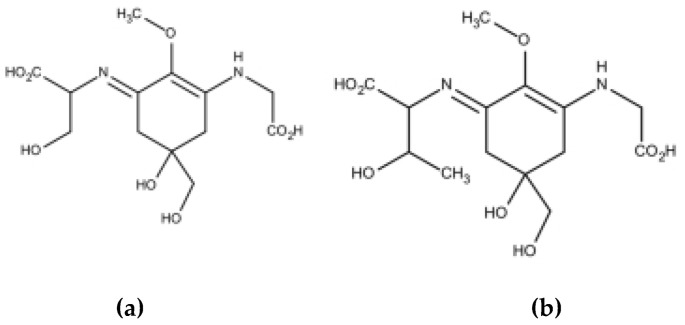
Chemical structure of Shinorine (**a**) and porphyre-334 (**b**).

**Figure 2 marinedrugs-18-00322-f002:**
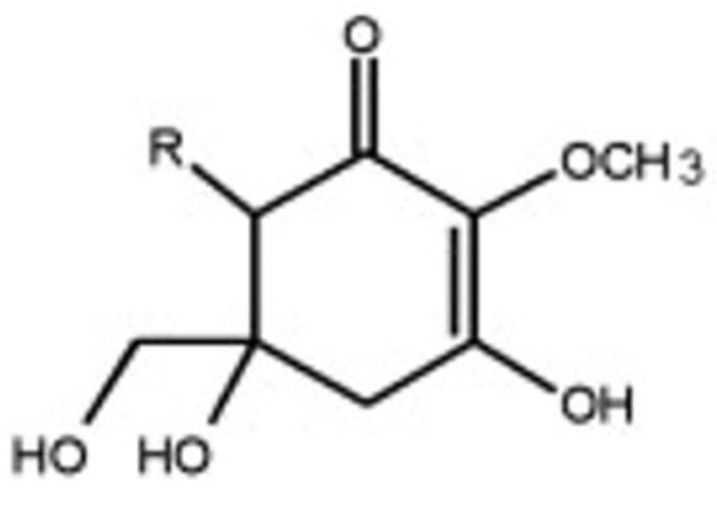
Chemical structure of gadusol.

**Figure 3 marinedrugs-18-00322-f003:**
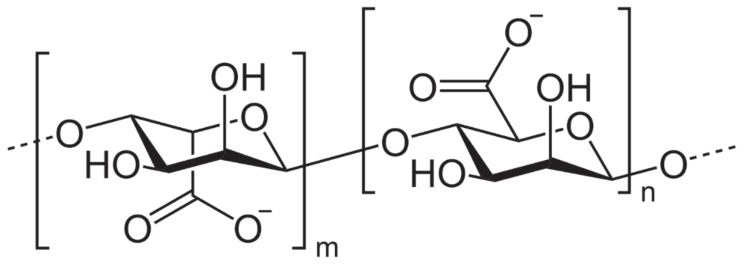
Chemical structure of alginates

**Figure 4 marinedrugs-18-00322-f004:**
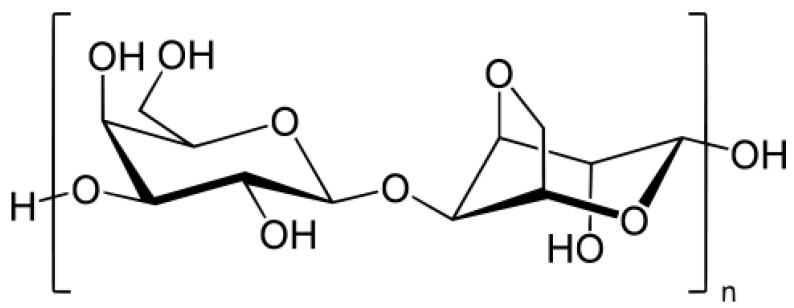
Chemical structure of a carrageenan.

**Figure 5 marinedrugs-18-00322-f005:**
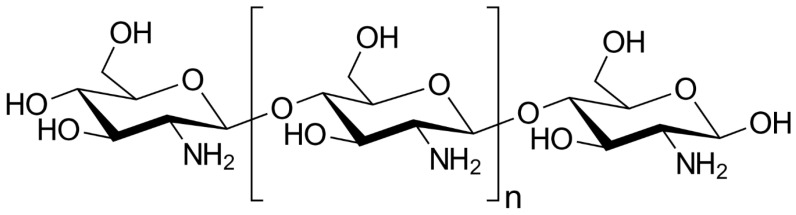
Chemical structure of chitosan.

**Figure 6 marinedrugs-18-00322-f006:**
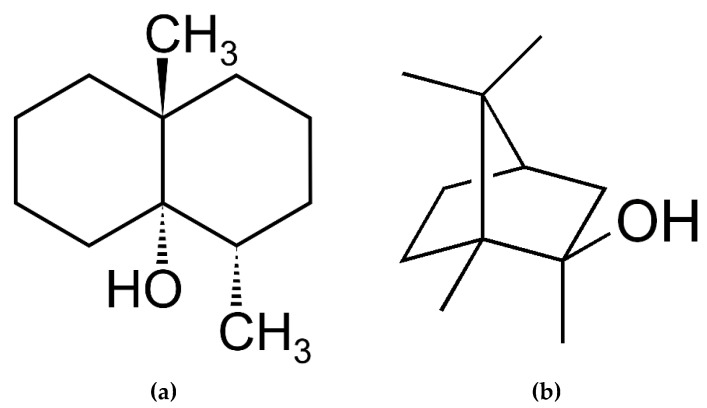
Chemical structure of geosmin (**a**) and 2-methylisoborneol (**b**).

**Figure 7 marinedrugs-18-00322-f007:**
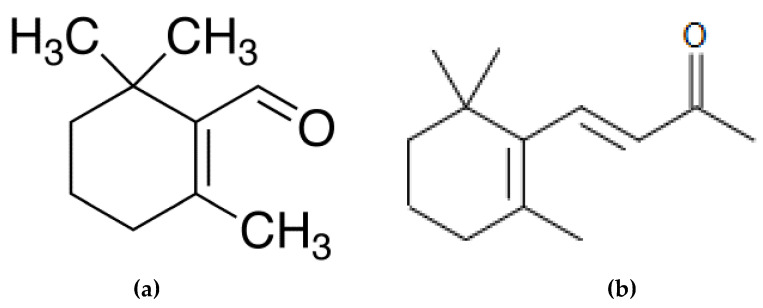
Chemical structure of bêta-cyclocitral (**a**) and bêta-ionone (**b**).

**Figure 8 marinedrugs-18-00322-f008:**
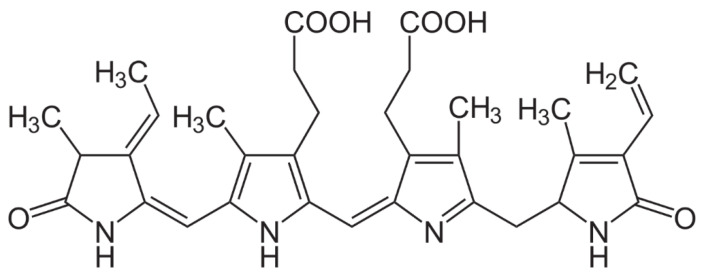
Chemical structure of R-phycoérythrine.

**Figure 9 marinedrugs-18-00322-f009:**
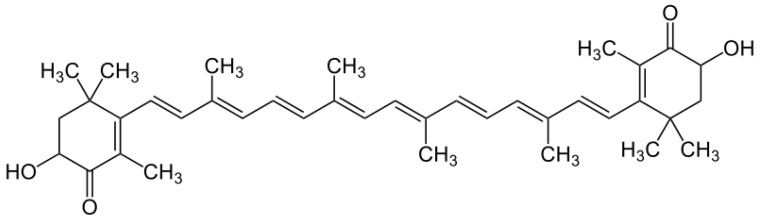
Chemical structure of astaxanthin.

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
