# Peer review of "Phycocosmetics and Other Marine Cosmetics, Specific Cosmetics Formulated Using Marine Resources"

_marinedrugs, 2020, doi:10.3390/md18060322_

Round 1

Reviewer 1 Report

This review is about the phycocosmetics based on the galenic formula. The topic is interesting and suitable for Marine Drugs. However, it cannot be published without revision of the following inquiries.

  1. The title is “Marine cosmetics” but there is a content of phycocosmetics. So, the title should be changed. There are huge classes of marine resources. The authors only pointed out some of them.
  2. The authors used galenic formula of excipients, active substances and additives. However, it is difficult to follow the logics. How about to use the “base + active ingredient + additives (surfactant or humectant)” instead. Base is O/W or W/O, and alginate and other polysaccharides could be additives. Some phycochemicals are to be active ingredients.
  3. All the figures are trivial; no new chemical structure at all.
  4. For active ingredients, the description should be target oriented; moisturizing, whitening, and UV-protection, so on. Or chemical-oriented; carbohydrate, polyphenol, and other chemicals. The present manuscript mixed up the two logics and is hard to read and understand the nature of phycocosmetics.
  5. Safety issues are to be included. There are so many heavy metals are incorporated in marine algae; arsenic, lead and others.

In summary, this manuscript is to be revised comprehensively before the acceptance.

Author Response

  1. The title has been changed.
  2. Not all cosmetics are emulsions. There are solutions (shampoos), suspensions (toothpaste, lipsticks...) Alginates and carrageenans are additives. The paragraph about them was included in the section 2.3 The marine world, a source of additives
  3. The figures are well known but illustrate the text.
  4. It is difficult to classify substances by chemical class because the same species may contain several substances of interest (for example : fatty acids and polyphenols). It is possible to classify the chemicals by their structure, uses, physical properties... We have chosen to classify them by cosmetic properties (hydration, anti-ageing, slimming...)
  5. At the end we added data on organic pollutants, metals and radionuclides.

Reviewer 2 Report

Introduction

Plant resources are also a very huge resource that can be used as an ingredient in cosmetics. The characteristics of the marine and plant resources should be compared and the advantages of the marine resources should be described.

2.2.3.

The description of this paragraph is too short. The names of the species of brown algae and their corresponding biological activities should be described.

Author Response

Introduction

The advantage of marine resources lies in the originality of the structures (phycobiliproteines, MAAs) A sentence has been added.

Section 2.2.3 was completed with the species name and activity type.

Round 2

Reviewer 1 Report

All the comments are cleared. The manuscript is accepted for publication.